# Efficacy and safety of single-step transepithelial photorefractive keratectomy with the all-surface laser ablation SCHWIND platform without mitomycin-C for high myopia: A retrospective study of 69 eyes

Jean Baptiste Giral[1], Florian Bloch[1], Maxime Sot[1], Yinka Zevering🄳[1], Arpine El Nar[2], Jean Charles Vermion[1], Christophe Goetz[2], Louis Lhuillier[1], Jean-Marc Perone[1]*

1 Department of Ophthalmology, Metz-Thionville Regional Hospital Center, University of Lorraine, Mercy Hospital, Metz, France, 2 Clinical Research Support Unit, Metz-Thionville Regional Hospital Center, University of Lorraine, Mercy Hospital, Metz, France

* jm.perone@chr-metz-thionville.fr

**Data Availability Statement:** The database file that was used for this paper are available from Zenodo

## Abstract

### Background

Studies suggest that transepithelial photorefractive keratectomy (TransPRK) with the all-surface laser ablation (ASLA)-SCHWIND platform is effective and safe for both low-moderate myopia and high myopia. In most studies, mitomycin-C is administered immediately after surgery to prevent corneal opacification (haze), which is a significant complication of photorefractive keratectomy in general. However, there is evidence that adjuvant mitomycin-C induces endothelial cytotoxicity. Moreover, a recent study showed that omitting adjuvant mitomycin-C did not increase haze in low-moderate myopia. The present case-series study examined the efficacy, safety, and haze rates of eyes with high myopia that underwent ASLA-SCHWIND TransPRK without adjuvant mitomycin-C.

### Methods

All consecutive eyes with high myopia ($\leq$-6 D) that were treated in 2018–2020 with the SCHWIND Amaris 500E® TransPRK excimer laser without adjuvant mitomycin-C in a tertiary-care hospital (France) and were followed up for 6 months were identified. Uncorrected visual acuity (UCVA), best spectacle-corrected visual acuity (BSCVA), and spherical equivalent (SE) were recorded before and after surgery. Postoperative haze was graded using the 4-grade Fantes scale. Efficacy rate (frequency of eyes with 6-month UCVA $\leq$0.1 logMAR), safety rate (frequency of eyes that lost <2 BSCVA lines), predictability (frequency of eyes with 6-month SE equal to target SE±0.5 D), efficacy index (mean UCVA at 6 months/ preoperative BSCVA), and safety index (BSCVA at 6 months/preoperative BSCVA) were computed.

DOI 10.5281/zenodo.5507429 (accessible on https://zenodo.org/record/5507430#.YUG6i7gzaUk).

**Funding:** The authors received no specific funding for this work.

**Competing interests:** The authors have declared that no competing interests exist.

## Results

Sixty-nine eyes (38 patients) were included. Mean preoperative and 6-month SE were -7.44 and -0.05 D, respectively. Mean 6-month UCVA and BSCVA were 0.00 and -0.02 logMAR, respectively. Efficacy rate and index were 95.7% and 1.08, respectively. Safety rate and index were 95.7% and 1.13, respectively. Predictability was 85.5%. Grade 3–4 haze never arose. At 6 months, the haze rate was zero.

## Conclusions

ASLA-SCHWIND TransPRK without mitomycin-C appears to be safe as well as effective and accurate for high myopia.

## Introduction

Photorefractive keratectomy (PRK) employs an excimer laser to remodel the corneal stroma after the corneal epithelium is removed down to Bowman's layer. Initially, it was a two-step procedure in which the corneal epithelium was first mechanically debrided or delaminated with alcohol [1]. This technique was developed in the late 1980s and was among the first refractive surgery techniques for low to moderate myopia [2]. While it was generally safe and effective, it also associated with several side effects, including significant postoperative pain, slow visual recovery, halo phenomena, and poor predictability, especially in cases of high myopia [3–5]. In particular, the wound-healing response caused by ablation of Bowman's layer and anterior stroma sometimes led to a significant long-term complication, namely, the development of corneal opacity (haze). The risk of this complication rose as myopia increased [6].

Due to these risks, conventional two-step PRK was quickly supplanted by laser in situ keratomileusis (LASIK), which emerged in the 1990s and generally has few of these side effects. In particular, haze is much rarer after LASIK [7–9]. However, LASIK treatment of high myopia still associates with some risk. This is partly because the larger flap size needed for high myopia cases can lead to healing complications [10]. Moreover, the deeper stromal ablation that is required can change corneal biomechanical stability and increase the risk of ectasia, which is a potentially devastating complication [11–13]. In addition, LASIK of high myopia cases carries an increased risk of rhegmatogenous retinal detachment that is secondary to the movement of the vitreous fluid during the relaxation of the suction circle [14].

A new option for myopia arose at the end of the 1990s, when Transepithelial PRK (TransPRK) appeared. In this single-step refractive technique, the excimer laser ablates both the corneal epithelium and stroma, thus eliminating the need to debride or delaminate the epithelium [1]. In 2007, SCHWIND eye-tech-solutions GmbH (Kleinostheim, Germany), a company that manufactures devices for ametropiae and corneal diseases, produced the all-surface laser ablation (ASLA) SCHWIND Amaris 500E TransPRK platform. This platform uses an aspheric nomogram to simultaneously ablate both the epithelium and stroma in one continuous step. Thus, the platform in radial phototherapeutic keratectomy (PTK) mode applies laser pulses centrally to peripherally in a defined-depth fashion. This ablates 55 μm of central tissue and about 65 μm of peripheral tissue (thus removing the epithelium) and delivers uniform energy over the entire corneal surface [1, 15, 16]. TransPRK, including SCHWIND TransPRK, is faster than conventional PRK since there is no contact between instruments and the eye. Multiple studies and a meta-analysis on these studies have also shown that in low-moderate

myopia, TransPRK is comparable to conventional PRK and LASIK in terms of efficacy, safety, and predictability; moreover, postoperative haze rates are lower than with conventional PRK [1, 16–24].

Seven recent studies have shown that TransPRK is also similarly or more effective than conventional PRK and LASIK in high myopia [24–30]. For example, the recent randomized trial of Mounir *et al.* showed that SCHWIND TransTRK and LASIK associated with similar efficacy (80.0% and 80.3%) and safety (95.0% and 95.1%) indices in 156 1:1-randomized patients [29]. Notably, in nearly all of these studies, the TransPRK protocol included routine treatment with mitomycin-C immediately after ablation. This procedure blocks keratocyte activation and proliferation and myofibroblast formation and therefore reduces the incidence of haze [6, 31, 32]. Its use after refractive surgery was introduced in 2000 and it remains a commonly applied technique [33]. However, a recent study by Adib-Moghaddam *et al.* in patients with mild-moderate myopia showed that eyes that underwent mitomycin-C treatment after SCHWIND TransPRK exhibited greater loss of endothelial cells at 12 months than the contralateral eyes that underwent TransPRK without mitomycin-C [34]. This association between adjuvant mitomycin-C treatment and endothelial cytotoxicity was also observed by an earlier study on conventional PRK in mild-moderate myopia [35] and by an experimental study in rabbits [36]. Notably, Adib-Moghaddam *et al.* also found that the mitomycin-C-treated and untreated contralateral eyes did not differ in haze rate or efficacy variables [34]. Thus, it is possible that in the era of ASLA SCHWIND TransPRK, adjuvant mitomycin-C treatment may no longer be needed, at least for mild-moderate myopia.

A recent prospective cohort study by Zhang *et al.* also suggests that mitomycin-C may also not be needed after ASLA SCHWIND TransPRK for high myopia: they showed that at 12 months, eyes that were treated with TransPRK without mitomycin-C were comparable to LASIK-treated eyes in terms of efficacy and safety and significantly better in terms of residual myopia and predictability; importantly, none of the TransPRK-treated eyes had more than trace haze at the last follow-up [25].

To verify these observations, we conducted a case-series study on eyes with high myopia that underwent ASLA SCHWIND TransPRK without adjuvant mitomycin-C treatment.

## Methods

### Ethics

This retrospective case-series study was approved by the Ethics Committee of the French Society of Ophthalmology (IRB 00008855 Société Française d'Ophtalmologie IRB#1). It was conducted according to the Declaration of Helsinki. All patients provided written informed consent to undergo the TransPRK procedure and gave verbal consent for the use of their anonymized data for research purposes. The consent procedure was conducted in accordance with the reference methodology MR-004 of the National Commission for Information Technology and Liberties of France (No. 588909 v1).

### Patient selection

All consecutive adult (≥18 years) patients who had stable (variation of ≤0.5 D in >1 year) high myopia (≤-6 D) with or without astigmatism <3 D and underwent TransPRK with the ASLA SCHWIND Amaris 500E platform (SCHWIND eye-tech-solutions GmbH, Kleinostheim, Germany) in the Metz-Thionville Regional Hospital Center (Metz, France) in January 1 2018–January 1 2020 were included in the study. Patients were excluded if they had abnormal corneal topography or manifest keratoconus, any pre-existing eye pathology, a

history of eye surgery, active inflammatory or infectious eye disease, dermatological disease, or systemic connective tissue diseases.

## Preoperative assessment

All patients received information about the possibility of refractive surgery after an interview and ophthalmic assessment. During the interview, the patient was asked about their age, general and ophthalmic history, contact lens use, profession, extraprofessional activities, and motivations and expectations regarding refractive surgery. The ophthalmic assessment included self-refraction, keratometry, and pulsed air tonometry with the NIDEK Tonoref III® device (NIDEK Co. LTD, Tokyo, Japan); measurement of uncorrected visual acuity (UCVA) and subjective refraction with measurement of the best spectacle corrected visual acuity (BSCVA) that was expressed as logarithm of the minimum angle of resolution (logMAR); slit lamp biomicroscopic examination of the anterior segment with fluorescein test (Fluoresceine 0.5%; SERB, Paris, France) and dilated eye fundus; Scheimpflug corneal topography (Sirius, SCHWIND eye-tech-solutions GmbH, Kleinostheim, Germany); and macular OCT with the NIDEK RS-3000 OCT Retina Scan Advance® (NIDEK Co. LTD, Tokyo, Japan). The patient was then informed verbally about the risks and benefits of TransPRK with the SCHWIND Amaris 500E platform and how the surgical procedure would be conducted. They were also given a pamphlet (SFO sheet no. 9A) that provided the same information. If they agreed to the procedure and were wearing soft or hard contact lenses, they were invited to stop wearing them for one or three week before the procedure, respectively.

## Operative procedure

Antibioprophylaxis treatment was started the day before surgery with 2 drops/day Azyter® (Azithromycin dihydrate 15 mg/g; Théa Pharma, Clermont-Ferrand, France) and continued for a total of 3 days. Premedication with Tranxene® (Clorazepate 10 mg; Sanofi-Aventis, Gentilly, France; one capsule the day before and two hours before the laser treatment) was also prescribed.

Ten minutes before the operation, topical anesthesia was provided by ocular instillation of one drop of oxybuprocaine 1.6 mg/0.4 mL (Théa Pharma, Clermont-Ferrand, France). Palpebral disinfection was performed according to the Betadine Scrub® (Povidone iodine 4%, Meda Pharma, Paris, France) protocol. Ocular disinfection was performed for 2 minutes with Betadine Ophthalmic® (Povidone iodine 5%; Meda Pharma, Paris, France) followed by rinsing for approximately 10 seconds with 50 ml cooled (4˚C) sterile physiological saline (0.9% NaCL; Laboratoire Gilbert, Hérouville Saint-Clair, France). Any surplus of physiological saline in the conjunctival fornices was dried by Eye Microsponges PVA 40–405 (Sanotek Laboratory, l'Hay les Roses, France) before the laser procedure.

TransPRK was then conducted in a single step with the SCHWIND Amaris 500E platform. The refractive treatment was calculated by using the ORK-CAM® software that was developed by SCHWIND; it provides an aspheric ablation profile that minimizes induced optical aberrations and optimizes contrast and quality of vision.

As soon as the laser treatment was completed, the cornea was cooled by rinsing for approximately 10 seconds with 50 ml cooled (4˚C) sterile physiological saline (0.9% NaCL; Laboratoire Gilbert, Hérouville Saint-Clair, France). A drop of Tobrex® (Tobramycin 0.3%; Novartis Pharma SAS, Rueil-Malmaison, France) and a lubricant (Celluvisc ®, carmellose sodium; Allergan, Courbevoie, France) were then instilled. A dressing lens (Air Optix Night & Day Aqua®; Laboratoire Alcon, Rueil-Malmaison, France) was then placed for 48 hours. Patients

were advised to wear sunglasses at all times for the first 3 days to prevent photophobia and thereafter when outside for the first 3 months.

## Postoperative follow-up

Postoperative follow-up visits were scheduled at 1 week and 1, 3, and 6 months after surgery. The immediate postoperative treatment was 2 drops/day Azyter® (Azithromycin; Théa Pharma, Clermont-Ferrand, France) for 2 days and 4 drops/day Vitamin A eye drops (Rétinol 1500 UI/mL; Laboratoire Europhta, Monaco) until the first visit. In the event of pain, general analgesia with 2 capsules 4 times/day of Dafalgan codeine® 500 mg/30 mg (Paracetamol-Codeine; UPSA SAS, Rueil-Malmaison, France) was also prescribed. In the event of severe pain, local analgesic treatment (instillation of one eye drop/4h of oxybuprocaine 1.6 mg/0.4 mL; Théa Pharma, Clermont-Ferrand, France) was prescribed for the first 48 hours. The dressing lenses were removed within 48 hours of the operation by the patient if he or she had worn lenses previously or by an ophthalmologist if not. Additional treatment with an ophthalmic vitamin A ointment (Retinol 250 UI/100 g; Allergan, Courbevoie, France) that was used that evening and in subsequent evenings was then started for 1 month.

During each follow-up visit, the following assessments were made: monocular then binocular UCVA (expressed as logMAR); subjective refraction with corrective lenses and measurement of BSCVA; and slit lamp examination with a fluorescein test and use of the Fantes scale [37] to grade corneal haze from 0 (absence of corneal haze) to 4 (complete opacification of the stroma preventing access to the anterior chamber) (see Table 3).

If the fluorescein test did not detect an epithelial defect during the first follow-up visit, treatment with 3 drops/day Flucon® (Fluorometholone 0.1%; Novartis Pharma SAS, Rueil-Malmaison, France) was introduced along with lubricating treatment with 3 drops/day Vismed Unidose® (Sodium Hyaluronate; Horus Pharma, Saint-Laurent-du-Var, France). The Flucon® treatment was gradually reduced during the 3 months of follow-up to 2 instillations/day in the second month and then 1 instillation/day in the third month. If the patient did exhibit an epithelial defect at the first visit, he or she was monitored weekly until epithelial healing was completed. Flucon® and Vismed Unidose® were then introduced. Note that no steroidal or non-steroidal anti-inflammatory treatment was introduced before the first follow-up visit.

## Study outcome variables

The UCVA, BSCVA, spherical equivalent, and astigmatism of each patient before and at the three follow-up visits were recorded. Corneal haze was graded at the three postoperative visits.

The primary outcome variables of the study were TransPRK efficacy, safety, and Fantes grade haze during follow-up. TransPRK was considered effective for individual operated eyes when their UCVA was $\leq 0.1$ logMAR (or $\geq 8/10$ on the decimal scale) at 6 months. TransPRK was considered to be effective for the whole cohort if $\geq 80\%$ patients achieved this threshold. The efficacy index was calculated as the ratio of mean UCVA at 6 months to mean preoperative BSCVA. TransPRK was considered safe for individual operated eyes if they lost $<2$ lines of BSCVA at 6 months relative to the preoperative BSCVA. TransPRK was considered to be safe for the whole cohort if $\geq 80\%$ of the patients achieved this threshold. The safety index was calculated as the ratio of mean BSCVA at 6 months to mean preoperative BSCVA. Efficacy and safety indices of $\geq 1.0$ are considered to reflect good outcomes [1].

Secondary outcome variables were: mean residual spherical equivalent at 6 months; frequency of eyes whose spherical equivalent at 6 months was the target spherical equivalent ± 0.5 D or ± 1 D; predictability of the SCHWIND TransPRK nomogram, defined as the frequency of eyes whose change in spherical refraction at 6 months matched the target refraction ± 0.5 D

at 6 months; residual astigmatism at 6 months (mean and frequency with residual astigmatism <0.25 D); and non-haze complications during follow-up. Note that predictability values close to 100% indicate excellent accuracy of the TransPRK [1, 30].

## Statistical analyses

The data were complete for all patients. All continuous data were expressed as mean ± standard deviations (range). Categorical data were expressed as *n* (%). The relationship between the spherical equivalent at 6 months and the theoretical expected spherical equivalent after TransPRK was assessed by linear regression analysis. All analyses were performed with the SAS 9.4 software package (SAS Inst., Cary, NC).

## Results

### Study population and surgery

The study population consisted of 69 high myopic eyes from 38 patients. Before the procedure, the mean myopia, astigmatism, and spherical equivalent of the study eyes were -7.44, -0.84, and -7.84 D, respectively. Preoperative UCVA and BSCVA were 1.28 and 0.03 logMAR, respectively. The patients were 32 years old on average and 62% were women. Of the 69 eyes, 87% had previously worn contact lenses (Table 1).

The mean TransPRK optical zone, transition zone, and total ablation thickness were 6.07 mm, 1.88 mm, and 157.42 μm, respectively (Table 1). All patients completed 6 months of follow-up.

### Efficacy, safety, and corneal haze associated with SCHWIND TransPRK

Six months after surgery, the mean UCVA was 0.00 logMAR (9/10 on the decimal scale) and 95.6% (*n* = 66) of the eyes achieved UCVA ≤0.1 logMAR (or ≥8/10) (Fig 1 and Table 2). The efficacy index was 1.08 ± 0.18 (Table 2).

**Table 1. Baseline characteristics and surgical variables in 38 patients (69 eyes).**

| Characteristics | Values |
|---|---|
| Age, years | 32 ± 4 (25–43) |
| Sex ratio M/W (%M) | 13/25 (34%) |
| Myopia, D | -7.44 ±1.49 (-12.0 –-6.0) |
| Astigmatism, D | -0.84 ± 0.65 (-2.75–0) |
| Spherical equivalence, D | -7.84 ± 1.59 (-12.5 –-6.0) |
| UCVA, logMAR | 1.28 ± 0.07 (1.3–0.9) |
| BSCVA, logMAR | 0.03 ± 0.03 (-0.1–0.1) |
| K1, D | 43.47 ± 1.38 (40.9–46.65) |
| K2, D | 44.67 ± 1.41 (42.02–48.24) |
| Mean K, D | 44.07 ± 1.35 (41.46–47.15) |
| CCT, μm | 555.22 ± 25.27 (515–610) |
| Optical zone, mm | 6.07 ± 0.31 (5.3–6.3) |
| Transition zone, mm | 1.88 ± 0.40 (1.6–2.0) |
| Total ablation thickness, μm | 157.42 ± 14.76 (129.22–192.24) |

The data are expressed as mean ± standard deviation (range) or *n* (%).

BSCVA, best spectacle corrected visual acuity; CCT, central corneal thickness, K, keratometry; M, men; W, women.

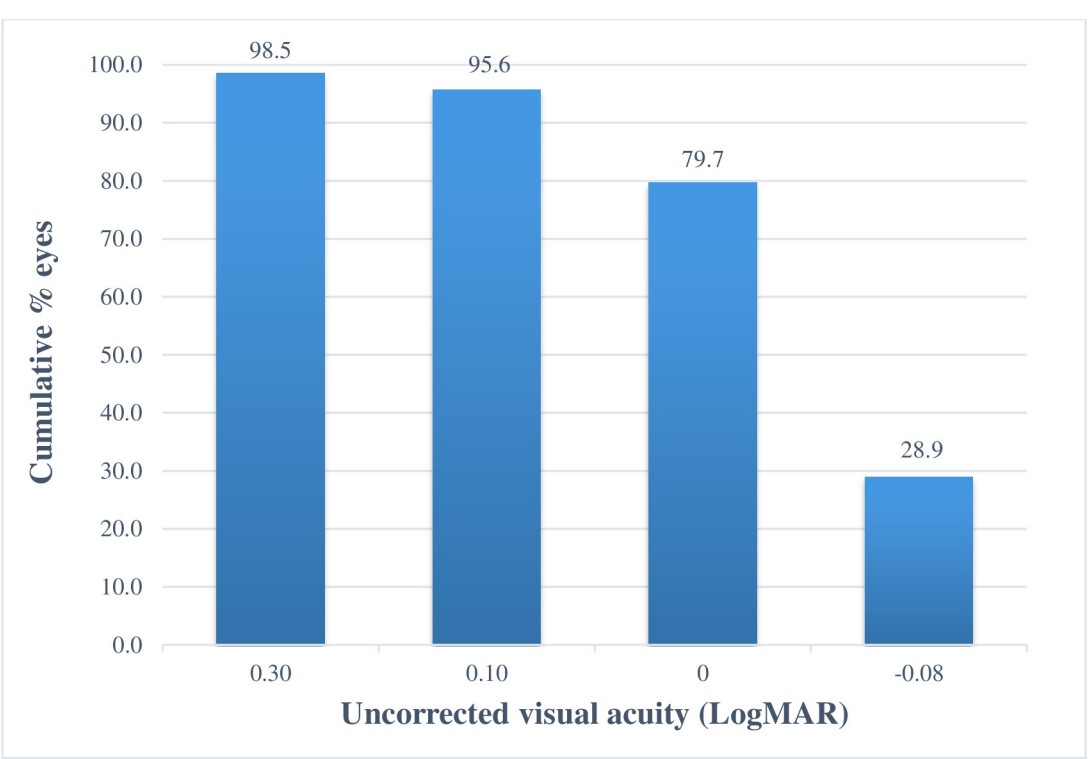

**Fig 1. Cumulative uncorrected visual acuity at 6 months follow-up (*n* = 69).**

In total, four eyes (5.8%) lost $\geq$2 lines of BSCVA: three and one eyes lost 2 and 3 lines, respectively. The remaining 65 eyes (94.2%) lost <2 lines: 35 eyes had the same BSCVA, 22 eyes gained one line, and two eyes gained two lines (Fig 2). The safety index was 1.13 ± 0.14 (Table 2).

**Table 2. Postoperative results (*n* = 69 eyes).**

| Variables | Follow-up visit | | | |
|---|---|---|---|---|
| | **Week 1** | **Month 1** | **Month 3** | **Month 6** |
| UCVA, LogMAR | 0.11 ± 0.12 | 0.07 ± 0.21 | 0.05 ± 0.23 | 0.00 ± 0.09 |
| VA ≤ 0.1 LogMAR, % (eyes) | - | - | 95.7 (66) | 95.7 (66) |
| VA ≤ 0 LogMAR,% (eyes) | - | - | 79.7 (55) | 81.2 (56) |
| BSCVA, logMAR | - | 0.04 ± 0.21 | 0.02 ± 0.21 | -0.02 ± 0.05 |
| Efficacy index | - | - | 1.07 ± 0.2 | 1.08 ± 0.18 |
| Lost 2 or more BSCVA lines, % (eyes) | - | - | 5.8 (4) | 4.3 (3) |
| Safety index | - | - | 1.13 ± 0.4 | 1.13 ± 0.14 |
| Sphere, D | - | 0.03 ± 0.54 | -0.03 ± 0.43 | -0.02 ±0.37 |
| Cylinder, D | - | -0.09 ± 0.29 | -0.07 ± 0.26 | -0.05±0.17 |
| Spherical equivalent, D | - | -0.02 ± 0.55 | -0.07 ± 0.50 | -0.05±0.42 |
| % eyes within ± 0.5 D of target SE | - | - | 81.2 (56) | 85.5 (59) |
| % eyes within ± 1.0 D of target SE | - | - | 95.7 (66) | 97.1 (67) |

The data are expressed as mean ± standard deviation or % (*n*).

BSCVA, best spectacle corrected visual acuity; SE, spherical equivalent; UCVA, uncorrected visual acuity.

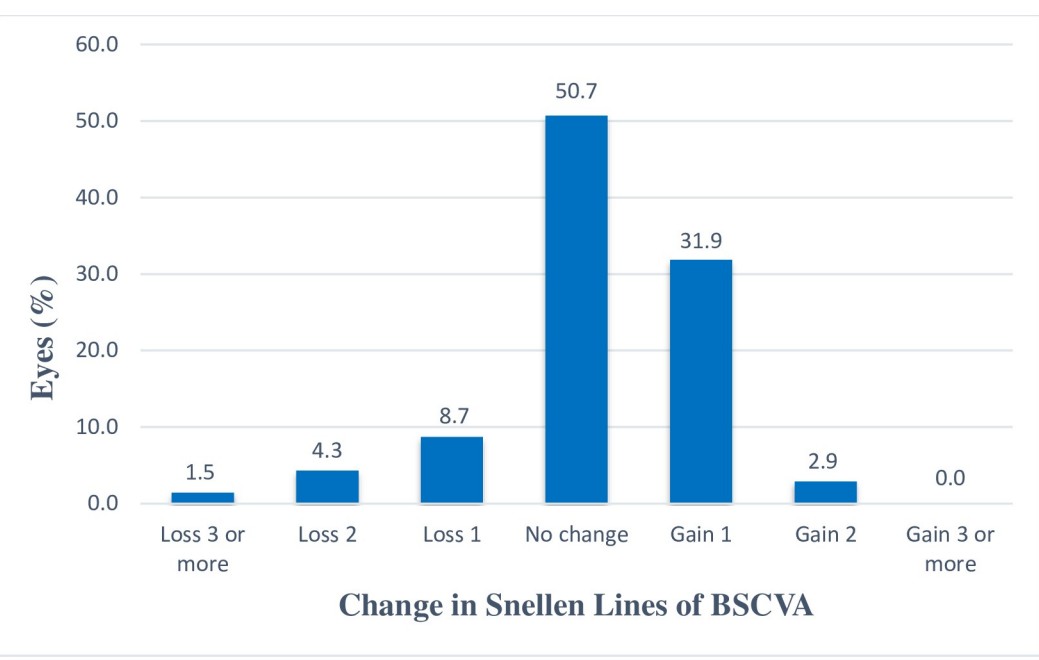

**Fig 2. Change in best spectacle corrected visual acuity lines at 6 months relative to baseline.**

Fantes grading of corneal haze during follow-up showed that at the first postoperative week, 23% and 7% of the eyes presented with grade 1 (corneal fog that does not prevent the visibility of details of the iris) and grade 2 (slight fading of details of the iris) corneal haze. Grade 3 and 4 haze was never observed. By the first month, only three eyes (4%) had corneal haze, and all were grade 1. At 3 and 6 months, there were no cases of corneal haze (Table 3).

### Refractive results and predictability of ASLA SCHWIND TransPRK

The mean 6-month spherical equivalent was -0.05 D. The final spherical equivalent met the target within ± 0.5 D and ± 1 D in 59 (85.5%) and 67 (97.1%) eyes, respectively (Table 2). Linear regression analysis of the actual and targeted change in final spherical equivalent yielded an $R^2$ coefficient of 0.965 and a slope of 1.0029 (Fig 3). Thus, the SCHWIND nomogram achieved highly accurate ablation.

At 6 months, 17 eyes (24.6% of the cohort) presented an overcorrection.

The mean postoperative refractive astigmatism was -0.05 D, and 63 eyes (91.3%) had residual astigmatism of <0.25 D (Fig 4).

### Non-haze complications

The postoperative course of all eyes was normal. Cases of postoperative infection were not observed. Delays in epithelial healing were not observed at the 1-week follow-up visit. Two eyes that lost 2 or 3 lines of visual acuity required surgical revision after 3 months of follow-up. The retreatment resulted in satisfactory outcomes (0.1 and 0 logMAR) and no corneal haze 6 months after revision surgery.

### Discussion

The present study assessed the refractive results of ASLA SCHWIND TransPRK for high myopia without mitomycin-C. The mean preoperative and 6-month spherical equivalent values

**Table 3. Fantes grade and corneal haze during follow-up.**

| Grade | Slit Lamp description |
|---|---|
| 0 | No haze, completely clear cornea |
| 1 | Haze not interfering with visibility of fine iris details (only seen by broad tangential illumination) |
| 2 | Mild obscuration of iris details (seen on direct focal illumination) |
| 3 | Moderate obscuration of the iris and lens |
| 4 | Complete opacification of the stroma in the area of the scar, anterior chamber is totally obscured |
| | *n* (%) |
| Week 1 | |
| Fantes Grade 0 | 48 (70) |
| Fantes Grade 1 | 16 (23) |
| Fantes Grade 2 | 5 (7) |
| Month 1 | |
| Fantes Grade 0 | 66 (96) |
| Fantes Grade 1 | 3 (4) |
| Fantes Grade 2 | 0 (0) |
| Month 3 Month 6 | |
| Fantes Grade 0 | 69 (100) |
| Fantes Grade 1 | 0 (0) |
| Fantes Grade 2 | 0 (0) |
| Month 6 | |
| Fantes Grade 0 | 69 (100) |
| Fantes Grade 1 | 0 (0) |
| Fantes Grade 2 | 0 (0) |

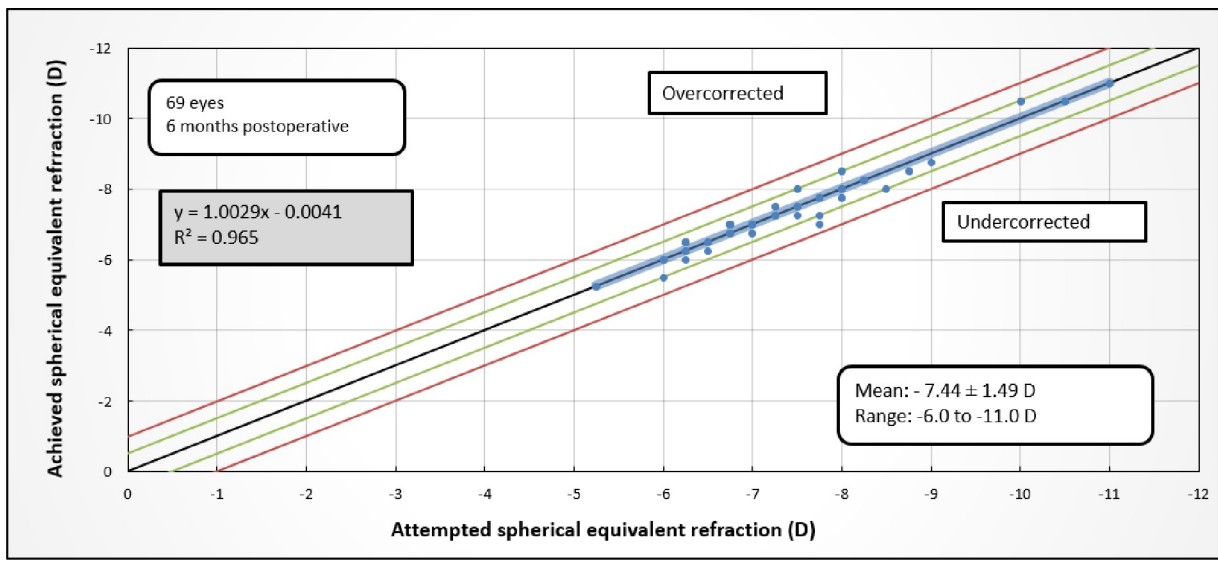

**Fig 3. Relationship between change in spherical equivalent at 6 months follow-up (achieved refraction) and the change in spherical equivalent that was targeted by the TransPRK platform (attempted refraction), as determined by linear regression.** The mean standard deviation and range spherical equivalent of the patients before surgery is shown in the bottom right panel. The green and red lines indicate correction to ±0.5 and ±1 D, respectively.

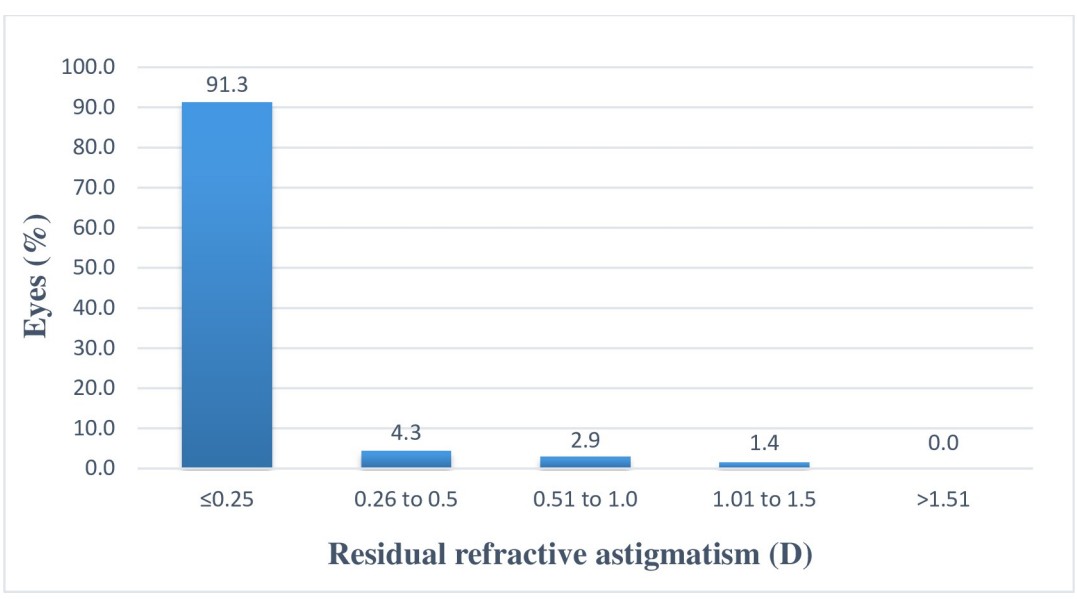

**Fig 4. Residual astigmatism at 3 months.**

were -7.44 and -0.05 D, respectively. The efficacy, safety, and predictability of this technique were all >94% and the efficacy and safety indices exceeded 1. Corneal haze never exceeded grade 2 and all corneal haze cases resolved completely by 3 months.

To our knowledge, seven studies have evaluated TransPRK for treating high myopia. The details of these studies are presented in Table 4. Nearly all TransPRK procedures were conducted with the SCHWIND platform; the exception was the study by Mounir *et al.*, who used a VISX S4IR excimer laser. Two studies were case series; in one, TransPRK was compared to refraction-matched historical cohorts. There was one randomized trial comparing TransPRK to LASIK. The remaining four studies were cohort studies that compared TransPRK to conventional PRK or LASIK. The follow-up durations were 3.5 or 12 months and mean preoperative myopia was generally less than -7 D. In the comparative studies, TransPRK was either better than or comparable to conventional PRK or LASIK in terms of efficacy, safety, residual spherical equivalent, and predictability. Our case series resembled the other TransPRK cohorts in terms of preoperative myopia and postoperative UCVA and efficacy and safety indices. Our predictability index (96%) tended to be higher than those of the other TransPRK cohorts (59–87%), possibly because refraction stabilizes at 6–12 months; our study duration was 6 months whereas the studies that reported predictability had durations of 12 months [25, 26, 28–30, 34]. Our low rate of clinically significant haze was also observed by six of the studies (0–1% of patients had Fantes grade ≥2 at the end of follow-up). In the remaining study (8% haze rate), patients were deemed to have haze if clinically significant haze had emerged at any time point during the 12-month study; it is likely that most, if not all, of these cases resolved by the end of follow-up.

In six of these seven studies, TransPRK was immediately succeeded by adjuvant mitomycin-C treatment. The exception was the prospective cohort study of Zhang *et al*, who, like us, did not use mitomycin-C. Similar to us, Zhang *et al.* had no cases of haze that exceeded grade 1 at the 12-month follow-up visit. It should be noted that the benefits of mitomycin-C treatment have not been re-examined since the introduction of the new excimer laser: the studies showing that mitomycin-C treatment leads to better PRK results in high myopia date back to more than ten years [32, 37, 38]. Notably, our treatment protocol involved the application of cooled (4˚C) physiological saline before and after refractive treatment. Similarly, Zhang *et al.*

**Table 4. Summary of studies on TransPRK in high myopia.**

| | | Aslanides 2014 Ref [16] | Adib 2017 Ref [27] | Antonios 2017 Ref [28] | Gershoni 2018 Ref [26] | Gadde 2020 Ref [24] | Zhang 2020 Ref [25] | Mounir 2020 Ref [29] | Our study 2021 |
|---|---|---|---|---|---|---|---|---|---|
| **Study Design** | | Case series* | Case series | Retro. cohort | Retro. cohort | Retro. Case-control | Prosp. cohort | Randomized trial | Case series |
| **Mitomycin-C use** | | **Y** | **Y** | **Y** | **Y** | **Y** | **NO** | **Y** | **NO** |
| Follow-up | | 12 months | 12 months | 12 months | 12 months | 3.5 months | 12 months | 12 months | 6 months |
| No. of eyes | TPRK | 41 | 30 | 59 | 674 | 23 | 85 | 72 | 69 |
| | Conv. PRK | 29 | - | 59 | - | 8 | - | - | - |
| | LASIK | 31 | - | - | 118 | - | 80 | 84 | - |
| Average myopia, D | TPRK | -7.89 | -6.72 | -7.24 | -7.45 [a] | X | -7.04 | -7.50 | -7.44 |
| | Conv. PRK | -8.25 | - | -7.53 | - | X | - | - | - |
| | LASIK | -7.41 | - | - | -6.73 [a] | - | -7.09 | -7.88 | - |
| UDVA, LogMAR (at end of FU) | TPRK | 0.00 [a] | -0.08 | 0.01 | 0.1 | 91%≤0.0 | -0.04 [a] | 0.3 | 0.05 |
| | Conv. PRK | 0.06 [a] | - | 0.006 | - | 100%≤0.0 | - | - | - |
| | LASIK | 0.05 [b] | - | - | 0.05 | - | -0.01 [a] | 0.3 | - |
| Efficacy Index | TPRK | X | 1.03 | 1.07 | 0.92 | 1.0 | 1.06 | 0.80 | 1.1 |
| | Conv. PRK | X | - | 1.09 | - | 0.99 | - | - | - |
| | LASIK | X | - | - | 0.95 | - | 1.01 | 0.83 | - |
| Safety Index | TPRK | X | X | 1.08 | 0.95 | 0.99 | 1.10 | 0.95 | 1.1 |
| | Conv. PRK | X | - | 1.10 | - | 0.99 | - | - | - |
| | LASIK | X | - | - | 0.97 | - | 1.08 | 0.95 | - |
| Residual SE, D | TPRK | -0.10 | -0.11 | 0.07 | 0.24 | X | -0.05 [a] | -0.65 | -0.07 |
| | Conv. PRK | -0.20 | - | -0.02 | - | X | - | - | - |
| | LASIK | -0.08 | - | - | 0.28 | - | -0.26 [a] | -0.69 | - |
| Predictability, %** | TPRK | 91 | 80 | 81 | 59 | X | 87 [a] | 71† | 96 |
| | Conv. PRK | 86 | - | 73 | - | X | - | - | - |
| | LASIK | 84 | - | - | 65 | - | 73 [a] | 86† | - |
| Haze ≥grade 2, % | TPRK | 0 | 0 | 0 | 8‡ | 1 [a] | 0 | 1 | 0 |
| | Conv. PRK | 0 | - | 0 | - | 0 [a] | - | - | - |
| | LASIK | 0 | - | - | - | - | - | - | - |

[a] TPRK differs significantly from LASIK in terms of the indicated variable.

[b] Conventional PRK differs significantly from TPRK in terms of the indicated variable.

Conv.PRK, conventional photorefractive keratectomy; FU, follow-up; LASIK, laser-assisted in situ keratomileusis; Mit, mitomycin-C; TPRK, Transepithelial photorefractive keratectomy; pros., prospective; retro., retrospective; SE, spherical equivalent; UCVA, uncorrected visual acuity; X, data not reported.

Black brackets show statistically significant differences between groups.

* This case series was compared to refraction-matched historical control eye groups that underwent conventional PRK or LASIK.

** Predictability is defined as a spherical equivalent refraction within 0.5 D of the target unless otherwise indicated.

† In Mounir et al., predictability was defined as spherical equivalent refraction within 1.00 D of the target.

‡ In Gershoni et al., a patient was considered to have haze if any ≥grade 2 haze emerged at any point during the 12-month study.

reported using cold balanced salt solution after laser ablation [25]. Two, non mutually exclusive, hypotheses can explain the lack of haze after TransPRK in high myopia without mitomycin-C. First, applying cooled physiological fluids before/after the procedure may reduce the increased corneal temperature caused by the thermal effect of excimer laser ablation [39]. Second, the smart laser beam pulse technology of the platform, which gently scans the corneal surface, optimizes the focal energy that is delivered and thereby results in less haze-inducing injury [21, 39, 40].

Transepithelial refractive surgery has been criticized for not accounting for potential individual variations in epithelial thickness, which could theoretically affect refractive results [20, 30]. The good outcomes of our cohort, and the other studies in myopia in general, do not support this notion [1, 16–30].

Our study had several limitations. First, it had a retrospective design and lacked a direct comparator group. Second, it had a short follow-up period (6 months), which means that we could not assess the myopia regression rate over time [27]. However, one of the case-series studies on TransPRK in high myopia found that refraction was stable 6 and 12 months after an initial further improvement at 3 months [27]; moreover, three of the comparative studies on TransPRK in high myopia showed that regression rates at 12 months were similar to those after conventional PRK and/or LASIK [16, 26, 29]. Third, corneal haze measurements were not conducted by the same physician during follow-up; this could have led to differences in diagnosis and grading and may have led to underestimation of the rate of haze. Finally, we did not objectively assess visual quality (aberrometry).

In conclusion, TransPRK appears to be a safe and effective procedure for high myopia. A low corneal haze rate was observed despite the fact that mitomycin-C was not applied postoperatively. Thus, mitomycin-C does not appear to be essential for refractive surgery for high myopia. However, further comparative studies with longer follow-up durations are needed to confirm this.

## Author Contributions

**Conceptualization:** Jean Baptiste Giral, Jean-Marc Perone.

**Data curation:** Jean Baptiste Giral, Florian Bloch, Maxime Sot, Jean Charles Vermion, Louis Lhuillier, Jean-Marc Perone.

**Formal analysis:** Jean Baptiste Giral, Christophe Goetz.

**Funding acquisition:** Christophe Goetz, Jean-Marc Perone.

**Investigation:** Jean Baptiste Giral, Florian Bloch, Maxime Sot, Jean Charles Vermion, Louis Lhuillier, Jean-Marc Perone.

**Methodology:** Jean Baptiste Giral, Christophe Goetz, Jean-Marc Perone.

**Project administration:** Christophe Goetz, Jean-Marc Perone.

**Resources:** Christophe Goetz, Jean-Marc Perone.

**Software:** Jean Baptiste Giral, Christophe Goetz, Jean-Marc Perone.

**Supervision:** Jean-Marc Perone.

**Validation:** Jean Baptiste Giral, Yinka Zevering, Jean-Marc Perone.

**Visualization:** Jean Baptiste Giral.

**Writing – original draft:** Jean Baptiste Giral.

**Writing – review & editing:** Jean Baptiste Giral, Yinka Zevering, Arpine El Nar, Jean-Marc Perone.

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
