## [Decision Letter · Decision Letter 0]

6 Aug 2021

PONE-D-21-13895

Efficacy and safety of single-step transepithelial photorefractive keratectomy with the all surface laser ablation SCHWIND platform without mitomycin-C for high myopia: a retrospective study of 69 eyes

PLOS ONE

Dear Dr. %Perone%,

Thank you for submitting your manuscript to PLOS ONE. After careful consideration, we feel that it has merit but does not fully meet PLOS ONE’s publication criteria as it currently stands. Therefore, we invite you to submit a revised version of the manuscript that addresses the points raised during the review process.

We look forward to receiving your revised manuscript.

Kind regards,

Rajiv R. Mohan, Ph.D.

Academic Editor

PLOS ONE

Journal Requirements:

2. Please provide additional details regarding participant consent for use of participant's anonymized data for research purposes. In the ethics statement in the Methods and online submission information, please describe whether verbal consent was informed, how verbal consent was documented and witnessed, and why written consent was not obtained. If your study included minors, state whether you obtained consent from parents or guardians.

3. We note that you have stated that you will provide repository information for your data at acceptance. Should your manuscript be accepted for publication, we will hold it until you provide the relevant accession numbers or DOIs necessary to access your data. If you wish to make changes to your Data Availability statement, please describe these changes in your cover letter and we will update your Data Availability statement to reflect the information you provide

Additional Editor Comments (if provided):

Dear Authors,

The reviews have identified some significant flaws that need to be addressed prior to making decision about publication. One of the reviewers feel that study design is poor required attention specially when it comes to Post PRK Haze. A minimum 6 months or more follow-up of patients is required. This has been identified a major flaw. Authors need to include additional data or provide solid evidence that it is not necessary. Other concerns expressed by reviewers are given below. Thank you for considering PLOS One journal for publishing your work.

Best wishes

Rajiv Mohan

Reviewers' comments:

Reviewer's Responses to Questions

**Comments to the Author**

1. Is the manuscript technically sound, and do the data support the conclusions?

Reviewer #1: Yes

Reviewer #2: Partly

2. Has the statistical analysis been performed appropriately and rigorously? 

Reviewer #1: Yes

Reviewer #2: No

3. Have the authors made all data underlying the findings in their manuscript fully available?

Reviewer #1: Yes

Reviewer #2: Yes

4. Is the manuscript presented in an intelligible fashion and written in standard English?

Reviewer #1: Yes

Reviewer #2: Yes

5. Review Comments to the Author

Reviewer #1: This is a meaning research about efficacy and safety of single-step transepithelial photorefractive keratectomy with the all surface laser ablation SCHWIND platform without mitomycin-C for high myopia. The major conclusion is suggested that ASLA-SCHWIND TransPRK without mitomycin-C appears to be safe as well as effective and accurate for high myopia. The research in general is meaningful. However, there are a number of issues that need to be attentioned.

In the part of case presentation:1.The detail of how to define the corneal haze measurementse is not described in the paper. I think it will be better to add the Fantes Grades standard.

2.In the paper, all consecutive eyes with high myopia (≤-6 D) that were treated in 2018–2020 with the 39 SCHWIND Amaris 500E® TransPRK excimer laser without adjuvant mitomycin-C in a 40 tertiary-care hospital (France) and were followed up for 3 months were identified. Is the range of high myopia(≤-6 D) correct in the method right?

3.Since some patients appear haze at 6 month, so it is really short of 3 months follow-up. So there could be a risk of false negatives in the results.

Reviewer #2: I congratulate the Authors on this wonderful study.

This is an important research question when it comes to Post PRK haze and Mitomycin- C.

However the Study design is poor and requires to be framed properly.

Also When it comes to Post PRK Haze, the minimum follow- up required is 6 months or above, which is lacking pre-dominantly in the manuscript

6. PLOS authors have the option to publish the peer review history of their article (what does this mean?). If published, this will include your full peer review and any attached files.

Reviewer #1: No

Reviewer #2: No

---

## [Author Response · Author response to Decision Letter 0]

17 Sep 2021

Response to Reviewers and Editors

Journal Requirements

We have reformatted the manuscript to meet PLOS One style requirements.

2. Please provide additional details regarding participant consent for use of participant's anonymized data for research purposes. In the ethics statement in the Methods and online submission information, please describe whether verbal consent was informed, how verbal consent was documented and witnessed, and why written consent was not obtained. If your study included minors, state whether you obtained consent from parents or guardians.

We included the following text to the Ethics section: “The consent procedure was conducted in accordance with the reference methodology MR-004 of the National Commission for Information Technology and Liberties of France (No. 588909 v1).”

3. We note that you have stated that you will provide repository information for your data at acceptance. Should your manuscript be accepted for publication, we will hold it until you provide the relevant accession numbers or DOIs necessary to access your data. If you wish to make changes to your Data Availability statement, please describe these changes in your cover letter and we will update your Data Availability statement to reflect the information you provide

The dataset has been deposited in Zenodo. The DOI is 10.5281/zenodo.5507429.

We have replaced the text “data not shown” with the visual acuity of the two patients who had to undergo surgical revision because they lost 2–3 lines of visual acuity after PRK: “The retreatment resulted in satisfactory outcomes (0.1 and 0 logMAR) and no corneal haze 6 months after revision surgery.”

Additional Editor Comments

The reviews have identified some significant flaws that need to be addressed prior to making decision about publication. One of the reviewers feel that study design is poor required attention specially when it comes to Post PRK Haze. A minimum 6 months or more follow-up of patients is required. This has been identified a major flaw. Authors need to include additional data or provide solid evidence that it is not necessary. Other concerns expressed by reviewers are given below. 

We have supplied the 6 month follow-up data for not only post-PRK haze but also other variables. As a result, Tables 2 and 3 and Figures 1–4 were revised along with the relevant text in the Abstract and Results section of the manuscript.

Post-PRK haze was also not observed at 6 months.

Reviewers' comments

Reviewer #1: This is a meaning research about efficacy and safety of single-step transepithelial photorefractive keratectomy with the all surface laser ablation SCHWIND platform without mitomycin-C for high myopia. The major conclusion is suggested that ASLA-SCHWIND TransPRK without mitomycin-C appears to be safe as well as effective and accurate for high myopia. The research in general is meaningful. However, there are a number of issues that need to be attentioned.

Thank you very much for your thoughtful consideration of our paper. We have addressed your comments to the best of our ability.

In the part of case presentation:1.The detail of how to define the corneal haze measurementse is not described in the paper. I think it will be better to add the Fantes Grades standard.

We added the Fantes Grade to Figure 3.

2.In the paper, all consecutive eyes with high myopia (≤-6 D) that were treated in 2018–2020 with the 39 SCHWIND Amaris 500E® TransPRK excimer laser without adjuvant mitomycin-C in a 40 tertiary-care hospital (France) and were followed up for 3 months were identified. Is the range of high myopia(≤-6 D) correct in the method right?

Yes, all eyes treated are < or = to -6D.

3.Since some patients appear haze at 6 month, so it is really short of 3 months follow-up. So there could be a risk of false negatives in the results.

We have supplied the 6 month follow-up data for not only post-PRK haze but also other variables. As a result, Tables 2 and 3 and Figures 1–4 were revised along with the relevant text in the Abstract and Results section of the manuscript.

Post-PRK haze was also not observed at 6 months.

Reviewer #2: I congratulate the Authors on this wonderful study.

Thank you very much for your thoughtful consideration of our paper. We have addressed your comments to the best of our ability.

This is an important research question when it comes to Post PRK haze and Mitomycin- C.

However the Study design is poor and requires to be framed properly.

Also When it comes to Post PRK Haze, the minimum follow- up required is 6 months or above, which is lacking pre-dominantly in the manuscript

We have supplied the 6 month follow-up data for not only post-PRK haze but also other variables. As a result, Tables 2 and 3 and Figures 1–4 were revised along with the relevant text in the Abstract and Results section of the manuscript.

Post-PRK haze was also not observed at 6 months.

---

## [Decision Letter · Decision Letter 1]

2 Nov 2021

Efficacy and safety of single-step transepithelial photorefractive keratectomy with the all surface laser ablation SCHWIND platform without mitomycin-C for high myopia: a retrospective study of 69 eyes

PONE-D-21-13895R1

Dear Dr. %Jean-Marc%,

We’re pleased to inform you that your manuscript has been judged scientifically suitable for publication and will be formally accepted for publication once it meets all outstanding technical requirements.

Kind regards,

Rajiv R. Mohan, Ph.D.

Academic Editor

PLOS ONE

Additional Editor Comments (optional):

Thanks for adequately addressing raised concerns

Reviewers' comments:

Reviewer's Responses to Questions

**Comments to the Author**

1. If the authors have adequately addressed your comments raised in a previous round of review and you feel that this manuscript is now acceptable for publication, you may indicate that here to bypass the “Comments to the Author” section, enter your conflict of interest statement in the “Confidential to Editor” section, and submit your "Accept" recommendation.

Reviewer #1: (No Response)

2. Is the manuscript technically sound, and do the data support the conclusions?

Reviewer #1: (No Response)

3. Has the statistical analysis been performed appropriately and rigorously? 

Reviewer #1: (No Response)

4. Have the authors made all data underlying the findings in their manuscript fully available?

Reviewer #1: (No Response)

5. Is the manuscript presented in an intelligible fashion and written in standard English?

Reviewer #1: (No Response)

6. Review Comments to the Author

Reviewer #1: (No Response)

---

## [Editor Report · Acceptance letter]

9 Nov 2021

PONE-D-21-13895R1 

Efficacy and safety of single-step transepithelial photorefractive keratectomy with the all-surface laser ablation SCHWIND platform without mitomycin-C for high myopia: a retrospective study of 69 eyes 

Dear Dr. Perone:

I'm pleased to inform you that your manuscript has been deemed suitable for publication in PLOS ONE. Congratulations! Your manuscript is now with our production department. 

Kind regards, 

on behalf of

Dr. Rajiv R. Mohan 

Academic Editor

PLOS ONE